# Heavy-Metal Phytoremediation from Livestock Wastewater and Exploitation of Exhausted Biomass

**DOI:** 10.3390/ijerph18052239

**Published:** 2021-02-24

**Authors:** Monika Hejna, Elisabetta Onelli, Alessandra Moscatelli, Maurizio Bellotto, Cinzia Cristiani, Nadia Stroppa, Luciana Rossi

**Affiliations:** 1Department of Health, Animal Science and Food Safety, Università degli Studi di Milano, 20133 Milan, Italy; monika.hejna@unimi.it (M.H.); luciana.rossi@unimi.it (L.R.); 2Department of Biosciences, Università degli Studi di Milano, 20133 Milan, Italy; alessandra.moscatelli@unimi.it (A.M.); nadia.stroppa@unimi.it (N.S.); 3Department of Chemistry, Materials and Chemical Engineering “G. Natta”, Politecnico di Milano, 20133 Milan, Italy; maurizio.bellotto@opigeo.eu (M.B.); cinzia.cristiani@polimi.it (C.C.)

**Keywords:** sustainable agriculture, phytoremediation techniques, heavy metals, wastewater, exhausted biomass reuse

## Abstract

Sustainable agriculture is aimed at long-term crop and livestock production with a minimal impact on the environment. However, agricultural practices from animal production can contribute to global pollution due to heavy metals from the feed additives that are used to ensure the nutritional requirements and also promote animal health and optimize production. The bioavailability of essential mineral sources is limited; thus, the metals are widely found in the manure. Via the manure, metallic ions can contaminate livestock wastewater, drastically reducing its potential recycling for irrigation. Phytoremediation, which is an efficient and cost-effective cleanup technique, could be implemented to reduce the wastewater pollution from livestock production, in order to maintain the water conservation. Plants use various strategies for the absorption and translocation of heavy metals, and they have been widely used to remediate livestock wastewater. In addition, the pollutants concentrated in the plants can be exhausted and used as heat to enhance plant growth and further concentrate the metals, making recycling a possible option. The biomass of the plants can also be used for biogas production in anaerobic fermentation. Combining phytoremediation and biorefinery processes would add value to both approaches and facilitate metal recovery. This review focuses on the concept of agro-ecology, specifically the excessive use of heavy metals in animal production, the various techniques and adaptations of the heavy-metal phytoremediation from livestock wastewater, and further applications of exhausted phytoremediated biomass.

## 1. Sustainability in Animal Production

Sustainability is aimed at the best use of environmental services without any negative or harmful impact [1]. Sustainable agriculture, which is focused on long-term crop and livestock production with a minimal impact on the environment, is thus an immediate global priority in order to ensure a balance between food production and the preservation of the environment. In addition, many goals related to sustainable agriculture and the modern principles of agro-ecology need to be effectively implemented in food production. These include (i) water conservation, (ii) a reduction in the use of fertilizers and pesticides, and (iii) promotion of biodiversity throughout the entire agro-ecosystem, as well as (iv) the continued economic profitability of farms [2,3]. Products, processes, and business models therefore need to be redesigned to maximize the value and utility of natural resources, while at the same time reducing adverse health and environmental impacts and climate changes [4].

Swine production is one of the most important branches of food production, and pork is the most consumed meat worldwide. The fast growth of the swine production sector has contributed to high economic gain due to the relatively short life cycle of pigs that have a high feed conversion ratio and reproductive rate. In animal production, nutrition, where animals are fed in line with the nutritional ecology strategy, and management are thus crucial in order to improve swine rearing, maintain animal well-being, meet the sustainable livestock production goals, and reduce water contamination, including heavy-metals (HMs) pollution from livestock-related activities [5].

## 2. The Importance of Heavy-Metals Use in Intensive Animal Production

Heavy metals (HMs) are metallic elements that have a high density compared to water and induce toxicity at low exposure levels [6,7,8,9,10]. Some heavy metals are essential to maintain biochemical and physiological functions, although excessive exposure has been linked with cellular or systemic disorders, acute and chronic toxicity, and sources of pollution [11]. Different HMs can enter animal diets both as contaminants/undesirable substances and as essential nutrients (Table 1) [12,13]. In the farming industry, essential trace elements are usually used as feed additives in order to not only satisfy the nutritional requirements and prevent nutritional deficiencies but also to promote health and welfare, optimize production, and improve food safety [14]. These elements are included within animal diets as mineral additives (Table 1) in compliance with the maximum admitted levels [15]. 

Heavy metals can also enter animal diets as contaminants with no established biological functions [16]. Arsenic (As), cadmium (Cd), chrome (Cr), lead (Pb), nickel (Ni), and mercury (Hg), which are a major risk to public health, are highly toxic and can induce organ damage even at low exposure levels. In line with previous research [17], swine and cattle were not affected by high amounts of these undesirable elements in the animal feed.

## 3. The Importance of Zinc and Copper as Alternative to Antibiotics in Animal Feeding

In intensive livestock, weaning is the most critical stage associated with low feed intake, influencing the growth performance, and with fluctuations in gut function, making piglets sensitive to digestive disorders. Weaning stress is a major cause of diarrhea and is often associated with many pathotypes of *Escherichia coli* infection of the intestine [18]. Previously, antibiotic growth promoters (AGPs) were used to reduce the instance of diarrhea at weaning. However, antibiotic resistance is a global concern, and restricting the use of antimicrobials in food-producing animals has reduced the prevalence of antimicrobial resistance in bacteria isolated from farm animals [19]. In the last decade in the EU, the state of livestock has thus changed significantly due to the ban on antibiotics [20], which has led to the study of alternative compounds. The first adopted alternative to feed antibiotics was the application of high doses of zinc and copper salts in the form of a premix to control enteric diseases in the growing phase. 

Zinc is toxic to animals, bacteria, and plants when encountered in high concentrations; however, it is also essential in the maintenance and restoration of barrier integrity, protection against pathogens, and modulation of the immune system by promoting antibody production against pathogens [11]. Today, zinc oxide (ZnO), which is the most common form of Zn, is widely used (up to 150 ppm in complete feed) to maintain the nutritional requirements of weaning [21]. In addition, Zn is applied in pharmacological doses (from 1000 to 3000 mg/kg feed) as an alternative to antibiotics in order to promote growth performance [22,23] and to control enteric intestinal bacterial disorders as well as enhancing the immune system for diarrhea prevention in pigs [22]. 

Copper (Cu) is also an important mineral that is widely used as a supplement in the diet of weaning pigs due to its role in increasing growth performance and favoring a better feed conversion ratio [24]. In pigs, dietary concentrations from 150 to 250 mg of Cu/kg can maximize growth performance without any risk of poisoning. The routine inclusion of CuSO_4,_ which is the most common form of Cu, in diets was found to reduce intestinal diseases and to be a cost-effective solution to the replacement of growth-promoting antimicrobials in pig diets [25]. 

## 4. Heavy Metals and Their Impact on the Environment

Despite the antibacterial and anti-inflammatory activities of zinc and copper salts, their wide use has raised many concerns related to environmental pollution, especially soil and groundwater contamination [6]. Mainly because the bioavailability and digestibility of Zn and Cu sources are limited, the metals are thus partially digested by animals, and the excess is eliminated by excretion in feces and found in the manure [26]. Several studies indicated that Zn and Cu are widely found in pig manure [17,27,28,29,30], cattle [30], and poultry livestock manure [27,28] as a result of their high doses in swine diets (Table 2). 

The HMs content in manure is therefore its reflection of the feed [27,31,32,33]. Through the manure, large amounts of metallic ions may enter to the livestock soil (Table 3) [6,7,28,34]. Moreover, through the animal manure, large amounts of metals may also enter to livestock wastewater and may drastically reducing their potential use for agricultural irrigation [27,28,29,30,35,36].

Moreover, the use of zinc and copper in animal feed may also have contributed to the emergence of methicillin-resistant *Staphylococcus aureus* (MRSA) due to the potential increase in the prevalence of antibiotic-resistant bacteria [11,37,38,39]. MRSA is rarely a clinically significant pathogen in pigs; however, it is one of leading causes of opportunistic infection in humans due to the increased burden on healthcare systems and treatment failures associated with antimicrobial therapy. Some recent evidence suggests that the use of zinc, copper, and metals in pigs is a risk factor for MRSA, as these compounds are associated with the co-selection of resistance genes to antibiotics [38]. Resistance determinants for zinc and copper are wide-spread among MRSA of pig origin and provide selective pressure on antimicrobial-resistant bacteria, which is why the implementation of high doses of these metals may play a role in maintaining antimicrobial resistance. In addition, copper can impose selective pressure on the bacterial community’s of its tolerance during manure composting. In light of this, exposure to trace metals may also contribute to antibiotic resistance, even in the absence of antibiotics themselves. Consequently, antibiotic resistance due to Zn and Cu may expose a zoonotic pathogen in animal production [11,37,38,40,41].

The anthropogenic contamination of the environment with HMs is thus a serious problem, and their long-term accumulation in the environment has led to their propagation in the food chain by accidental ingestion of soil, contamination of edible plants through the soil, or the consumption of contaminated animal-derived food products (Figure 1) [12].

The focus of EU environmental protection policies is on promoting economic growth together with the reduction in the impact of HMs [42]. Animals require essential minerals in their diets to meet the animals’ physiological needs and maintain various metabolic functions; hence, they are included in the European register of feed additives. European authorities have thus adopted various measures to control the HMs in the environment which are the result of human activities such as farming and industry. The comprehensive regulations on the maximum authorized admissible concentrations of essential and undesirable trace elements in additives have been established for animal nutrition [16,20]. The EU also recently decided to ban the inclusion of pharmacological levels of zinc oxide in animal feed after 2022 [43], because the overall balance between the benefits and risks remains negative for feed additives containing zinc oxide. Similarly, the new maximum admissible Cu content (for different Cu sources) was also established in complete feed for different animal species [44] in order to protect feed and food safety and ultimately human health.

## 5. How Can Plants Remove Metals from Livestock Wastewater?

Animals should be fed in accordance with nutritional ecology strategies because livestock nutrition plays a pivotal role in controlling environmental pollution [5]. However, if the nutritional ecology strategy is not sufficient to reduce the wastewater pollution from livestock production and to ensure water conservation, then efficient, cost-effective, reliable, and apt materials and methods need to be developed and locally implemented. This can be achieved through multidisciplinary research aimed at studying water pollution for the appropriate management of water resources [6,7], because in swine farms, HMs contamination of wastewater considerably reduces its potential for being recycled in irrigation [12,27,31,35]. Traditional wastewater treatment technologies [27] are ineffective in providing adequate safe water due to the increasing demand for water, coupled with stringent health guidelines and emerging contaminants. New materials and methods are therefore needed in order to obtain considerable potable water savings through the reuse of wastewater. 

Over the last few years, plants have been widely used to remediate wastewater from livestock induced pollution [45,46,47,48,49,50], due to their ability to remove HMs and other contaminants from the soil and water [51,52]. The most useful method for phytoremediation of livestock manure and wastewater can be achieved through constructed wetlands (CWs), which uptake metals and organic matter from water and mimic natural wetland processes at biological, chemical, and physiological levels [47,53,54,55]. CWs are mainly used for treating municipal, industrial, storm and agricultural waters, landfill leachate, and mine drainage wastewater, thus facilitating the recovery of both organic and inorganic compounds [53,54,55]. CWs can recover contaminants mainly due to the removal capability of microorganisms and to the pollutants’ adsorption from the substrate. Plants are able to extract contaminants through the root system and improve pollutant removal by providing an appropriate environment for rhizosphere microorganism growth or by modifying chemicals by improving their biological availability [51,52,53,54,55].

### 5.1. How Plants Function in the Phytoremediation of Heavy Metals

Some plants are able to uptake HMs from soil or water, due to the roots’ ability to adsorb and translocate these compounds in plant cells. Plants adopt both avoidance and tolerance to deal with the toxicity of HMs [52,56]. Avoidance is the first line of defense, and plants limit the uptake of HMs and their entry in the root tissues [57]. The mechanisms of avoidance work at different levels and involve (i) cell wall modification through callose, suberin, or lignin deposition [58]; (ii) the sequestration of metals into the cell wall [59,60,61,62,63]; (iii) the secretion of a root extracellular matrix which binds ions, stabilizing HMs in the rhizosphere and limiting their assimilation [57,64]; and (iv) the removal of excess metals by leaf glands [56]. Mycorrhizae could also function as avoidance mechanisms since fungi are able to uptake and immobilize metals into the mycelium, inhibiting translocation to the root tissues. In addition, fungi activate detoxification or chelation, thus reducing metal uptake from plants [65,66].

Rhizosphere microorganisms also support plants phytoremediation because they increase metal tolerance by enhancing metal bioavailability and their translocation in root tissues [67,68,69]. Tolerance mechanisms enable plant cells to accumulate metal ions in cell walls and vacuoles after chelation by amino acids, phytochelatins, metallothioneins, pectins, and phenols [70,71,72,73,74,75,76]. In addition, tolerance strategies involve proteins in metal detoxification metabolism, signal transduction, stress, and ROS signaling [77,78,79,80].

The ability to persist in HM-polluted environments enables some plants to be used for phytoremediation. Several phytoremediation strategies are applied for different substrates and different contaminants, most of which are used for both HMs-polluted soil and water. Phytostabilization uses plants to immobilize metals in the substrate or in the rhizosphere, preventing their leaching to groundwater. Microorganisms from the rhizosphere are also involved, which cooperate with plants, thus improving phytostabilization [52,56]. Phytoextraction, on the other hand, is exploited by plants to uptake metals inside the roots or underground organs, and to translocate and accumulate them in aboveground tissues. The evaporation of assimilated metals through leaves is defined as phytoevaporation [52,56]. 

For contaminated waters, the most common strategy used is phytofiltration, which includes the use of plant roots (rhizofiltration), shoots (caulofiltration), and seedlings (blastofiltration). In rhizofiltration, metals are adsorbed on the root or rhizome surface or accumulated in root or rhizome tissues. Most of the metals remain in the aboveground organs and only a small amount is translocated to the shoots. For this reason, plants with an extensive root system and high aboveground biomass are used [51,53,56,81]. This strategy, as well as phytoextraction and phytostabilization, is applied extensively in CWs.

### 5.2. CWs in the Phytoremediation of Heavy Metals from Livestock Wastewater

Pig manure is processed by separating the liquid and solid fractions. Waters obtained by sedimentation need to be refined in order to be reused for field irrigation. In the secondary or tertiary treatment, CWs are widely used, particularly for nitrogen and phosphorous recovery [45,53,82]. However, recent data have highlighted the use of CWs for HMs remediation [46,48,51,83,84,85]. In particular, both horizontal-vertical and surface-subsurface flow of CWs (S-CWs and SF-CWs, respectively) are used for treating HM-polluted water [46,54,86,87]. 

In the S-CW system, water flows above the substrate, while in SF-CWs, water flows inside the porous substrate (Figure 2) [53,54]. The S-CWs is effective for the removal of suspended solid, biochemical oxygen demand (BOD_5_), nitrogen, and HMs, while phosphorous removal is limited. The flow of water in SF-CWs can be horizontal or vertical. In the horizontal SF-CWs, the improvement in microorganism growth conditions in the rhizosphere enhances the removal of organic matter. On the other hand, in vertical SF-CWs, nitrogen and phosphorous removal also occurs. For this reason, the combination of different CWs are used in order to improve the efficiency of the remediation system [53,54]. More complex hybrid CWs were extensively described in Stefanakis et al. [54]. Usually, plants in the CWs accumulate HMs in their aboveground biomass [88,89]. This feature is considered important for a good bioremediation together with the limited translocation ability of HMs in the shoot and the tolerance to high level of HMs [90,91].

One of the most efficient plants used to reduce HMs in wetlands is the water hyacinth (*Eichhornia crassipes)*. This plant accumulates metals, such as Cd, Cr, Ni, Fe, and also Cu and Zn, in the root system and reduces their concentration in municipal and industrial wastewater, thus facilitating water reuse in agricultural systems [84,92,93]. *Eichhornia crassipes* is also efficient in CWs designed for pretreated swine effluent [45]. However, these SF-CWs are mostly able to efficiently remove suspended solids (96–99%), chemical oxygen demand (COD; 77–84%), total phosphorous (47–59%), and total nitrogen (10–24%). In terms of HMs, macrophytes have been found to be more effective for treating the liquid fraction of municipal wastewater or pig manure [46,85,86]. In fact, CWs with *Phragmites australis* reduce Cu and Zn levels, as well as COD, phosphorous, and nitrogen. However, while in these systems the sediment or belowground biomass plays a major role in Cu retention, plant uptake, and translocation accounts for about 30% of Zn retention [46]. A more recent study showed that *Phragmites australis* accumulates metals in the roots and rhizomes, and at lower levels, in the stems and leaves, and thus *Phragmites australis* efficiently removed Cu, Fe, Mn, and Zn from livestock wastewater in the CWs [48]. In wetland microcosms, also *Canna indica* L., *Typha angustifolia* L., and *Cyperus alternifolius* L. were very efficient for HMs removal in vertical CWs [89,94,95]. All these species with features described above fit for plants useful in CWs, because they showed (i) developed rhizomes and root system able to accumulate and retain HMs, (ii) low translocation ability of HMs in aboveground biomass, (iii) mechanisms of HMs tolerance, being able to growth in contaminated environment, and (iv) mechanisms able to growth in wetland environment [49,50].

While different publications documented the ability of plants in HMs remediation in CWs, only a few papers reported the biological mechanisms allowing plants to live in high-HMs-contaminated wet environment. In plants which are suitable for CWs, tolerance mechanisms include (i) synthesis of phytochelatins, peptides, and exudates to chelate-free metal ions, (ii) the increasing of antioxidant enzyme activities, and (iii) the sequestration of HMs in organs or subcellular compartments [96,97,98,99]. Among plants with those features, *Typha latifolia* is a macrophyte which accumulates HMs in its tissues and has therefore been used in the phytoremediation of wastewater for irrigation reuse [49,50,86]. *Typha latifolia* was found to accumulate Mn, B, Pb, Zn, and Cu in the root system. However, as *Phragmites australis*, it had a limited ability to translocate metals in the shoots. This plant therefore plays an important role in both rhizofiltration and phytostabilization in CWs [86]. 

*Typha latifolia* also works in smaller systems, such as nonintensive pig farms [49,50]. In a pilot system for the refinement of the liquid fraction of manure, *Typha latifolia*, associated with the marsh fern (*Thelypteris palustris)* was effective in the uptake of Zn and Cu [49]. In this system, the contribution to metal stabilization of both the substrate and rhizosphere was found to be relevant. Both plants contribute to the phytoremediation of metals. *Typha latifolia* activates various tolerance mechanisms, making it more suitable for the long-term phytodepuration of livestock wastewater. On the other hand, marsh fern was more sensitive but accumulated metals more efficiently within a short time. Thus, the use of both plants in this phytoremediation system was significantly effective [50]. Moreover, although neither plant showed macroscopical alterations, microscopical observations revealed that both rhizome and leaf morphology were affected by the exposure to Zn and Cu. *Thelypteris palustris* was more sensitive than *Typha latifolia*, because the modification in both the leaf and rhizome cells (cell shape, cell wall thickness, and pectin distribution) and carbohydrate metabolism indicated that the marsh fern was affected more by the presence of the two metals. On the other hand, the accumulation of amyloplasts instead of chloroplasts during leaf senescence in *Typha latifolia*, together with the decrease in starch in rhizomes, could be considered as part of a tolerance mechanism [50]. The altered carbohydrate metabolism in this plant [45,46] could be due to the requirement of soluble sugars, which is important in preserving biological molecules and membranes while a plant is in stressed condition [100].

In several plant phytoremediation models, one tolerance strategy is sequestration into the cell wall, because cell-wall polysaccharides play a major role in binding and accumulating HMs in order to remove them from protoplasts [62,101]. In these plants, the tolerance mechanism induces a thickening of the cell wall and pectin remodeling by modulating the degree of melthyl-esterification, thus affecting the ability of the cell wall to bind metals [60,62,102,103]. Unlike most plants used for phytoremediation, *Typha latifolia* and *Thelypteris palustris* showed a significant reduction in cell wall thickening in rhizomes after metal exposure [50]. In addition, changes in the balance between esterified and de-esterified pectins did not follow the phytoremediation model, suggesting that these modifications are part of a toxic response rather than a tolerance mechanism. These modifications were related to alteration of cytoskeleton protein induced by metals exposures. Proteomic analysis showed a decrease of both actin and microtubules. Actin filaments and microtubules are needed for secretion processes during cell wall building suggesting that these modifications are not part of a tolerance mechanism [50]. 

Plants used for metals uptake are numerous, and Sandoval et al. [104] summarized some natural and ornamental plants for phytoremediation in CWs. However, the use of *Typha latifolia* and *Thelypteris palustris* in a pilot system suggested that macrophytes could be useful in CWs for HMs remediation of livestock wastewater, thanks to their ability to produce higher underground biomass able to accumulate HMs without translocating them in belowground organs. Toleration or avoiding mechanisms allowed plants to grow and act in CWs for long time.

## 6. Plant Reuse after Phytoremediation

After a successful phytoremediation, every part of the plant contains heavy-metal pollutants due to the transport processes. Phytoremediation may thus result in potentially hazardous biomass. The downstream processing of the biomass is therefore an integral part of the remediation approach. Phytoremediation techniques have several advantages over physical or chemical processes for treating wastewater. Physical processes involve the precipitation of the HMs as insoluble salts or hydroxides, followed by flocculation and separation. Chemical processes involve HMs adsorption onto a substrate, which is subsequently regenerated by ion exchange or disposed by landfilling. 

Both approaches need a significant surface area for the installations, use of chemicals (for pH control, flocculation, adsorption, and regeneration), and energy for pumping and stirring. The quality of the landscape is preserved or even improved by phytoremediation, which also has a limited environmental impact given that it uses mostly solar energy. However, life-cycle assessments (LCAs) have shown that if the produced biomass is not enhanced, the sustainability of phytoremediation is questionable compared to landfilling [105]. 

During phytoremediation, the pollutants are concentrated in the plants, and ashing of the exhaust harvested plants further concentrates the metals, making recycling possible. Moreover, the recovery of the ashing-generated heat may be used for enhancing plant growth [106,107]. Alternatively, the plants biomass may be used for biogas production in anaerobic fermentation processes [108,109,110].

### 6.1. Incineration

Phytoremediation biomass can be treated thermochemically, through gasification, pyrolysis or combustion, achieving a valorization to provide fuel gas that can be used for electricity generation or to produce heat [111]. The volume of the ashes is substantially reduced compared to the volume of the biomass. The HMs content in processed ash from the thermochemical process is further concentrated compared to the original biomass. The process conditions need to be selected so that they minimize HMs volatilization and concentrate them in the solid ashes. Reuse of the ash or recovery of the HMs is cost-effective, avoiding the disposal cost for toxic materials. This thermal treatment is used when the volumes produced are sufficient to operate an efficient large-scale efficient combustion. Heat generated directly through combustion or from the fuel gas can be used to foster plant growth and for the supply of ancillary items of the phytoremediation plant. 

The resulting ash may be used as a pozzolanic addition to hydraulic binders in the formulation of composite Portland cement [112] and geopolymers [113]. This exploits the hydraulic activity of the calcined phytoliths as a source of reactive silica. To prevent the captured metals from leaching and being redispersed into the environment, they need to be fixed in the hardened hydrated structure. The fixation of the captured metals is more efficient for geopolymeric binders than Portland cement, due to the different pH of the interstitial solution and the different hydrated mineral phases [114], thus making geopolymers more attractive than Portland cement stabilization.

Metal enrichment in the ashes could also impact metal recovery and recycling of metals, particularly those included in the critical raw materials list [115]. Recovery has great potential when a limited number of metals are present in relatively high concentrations. Among others, some examples are Cu and Zn, present in swine wastewater, as a result of their addition to the animal feed for their antibacterial and anti-inflammatory activities. Another example is the recovery and recycling of the nickel, whose demand is increasing for the production of batteries, to replace cobalt both for political and environmental issues [116].

The recovery of HMs from phytoremediation biomass ashes can be performed through pyrometallurgical processes [117], however these energy-intensive treatments are not suited to the low volumes of ashes produced. A more suitable approach is the recovery of metals via solid–liquid adsorption and desorption processes [118]. This process is suitable for treating small volumes of ashes, which are first treated with a digestion process in order to solubilize the HMs, and then processed to adsorb the valuable metals on a properly designed solid. The adsorbed metals are then selectively desorbed and the solid regenerated [119].

### 6.2. Biotechnological Process

Fermentation of the exhaust biomass has demonstrated its potential for the degradation of lignocellulose to produce sugars and organic molecules of industrial interest. Anaerobic digestion refers to how organic materials are decomposed by microorganisms to produce biogas under anaerobic conditions [120]. The biogas mixture obtained contains on average 60% methane, which can be used as a substitute for fuel in boilers. As it has high N, P, and K contents, the associated liquid fraction can be used in agriculture [121]. 

Few studies have been conducted on the production of biogas by anaerobic fermentation from plant biomass used for the phytoremediation of industrial waste. The quantitative and qualitative increase in biogas generation from water hyacinths and water chestnuts grown in brass and electroplating industry effluent has been observed by Verma et al. [122]. The positive role of the waste stream, enhancing biogas production, is due to the presence of various pollutants that act as micronutrients for aquatic macrophytes/methanogens, especially at lower concentrations. Biomass grown in higher effluent concentrations severely reduced the methane content in the biogas owing to the methanogenesis inhibition caused by toxic effects due to the higher concentrations of metals. The production of biogas from plant biomass used for phytoremediation of a Cu-contaminated mine site was studied by Cao et al. [123]. In this case, 100 mg kg^−1^ Cu also promoted the anaerobic digestion and shortened the digestion times compared to the control group with a low Cu content. On the other hand, the presence of 500, 1000, and 5000 mg kg^−1^ Cu decreased cumulative biogas production by 12.5%, 14.9%, and 41.2%, respectively. Even higher Cu concentrations (>1000 mg kg^−1^) significantly hampered the anaerobic digestion of plants. 

Sotenko et al. [124] showed that nickel extracted from plants (*Sinapis alba* and *Helianthus annuus*) grown in contaminated soil can be easily extracted by aqueous extraction under mild conditions. The biomass was then subjected to solid-state fermentation as a downstream process. The plants that accumulated 11.9–15.1 ppm of nickel were degraded by the fungus *P. chrysosporium*. The contamination worsened the degradation of *H. annuus* by 10% but not that of *S. alba*. The pretreatment by aqueous extraction prior to fermentation increased the degradation yield by 14–15% for *S. alba*. Extraction was also found to significantly reduce the amount of soluble sugars from 56–106 to 18–24 mg g_dw_. This led to the deficiency of available sugars and phenols and to the enhancement of the degrading fungus growth for *S. alba* but not for *H. annuus*. The degradation of lignocellulose that underwent pretreatment led to a higher final amount of sugars (ca. 50 mg g_dw_) and phenols (5–6 mg g_dw_) in the extracts.

## 7. Discussion and Conclusions

This review has focused on animal production as a possible source of HMs in the water which have negative effects on human and animal health. The concept of agro-ecology has been highlighted by describing phytoremediation strategies for HMs recovery from livestock wastewater and by the reuse of exhausted phytoremediated biomass.

Agricultural activity is a significant global concern in terms of its negative impact on the environment and on food chain [17,42,125,126]. Animal production and pig livestock in particular are a key link in the food chain and in the spread of heavy metals. Arsenic, cadmium, chrome, lead, and mercury are considered priority hazards to public health due to their high toxicity even at low exposure levels [16]. However, in general, they are well controlled in the field. Conversely, many heavy metals (cobalt, copper, chromium, iron, manganese, molybdenum, selenium, zinc, and nickel) are essential nutrients with a wide array of vital physiological functions and which are usually added as additives in feed to satisfy the daily requirements [20]. Furthermore, in commercial conditions, feeding piglets with high doses of Zn and/or Cu stimulates piglets’ daily gain and decreases the feed conversion factor. Until now, Zn and Cu have been widely used as growth promoters, although Europe is now adopting strategies for their reduction. Considering the low bioavailability of the mineral additives, which are more concentrated in feces that are usually used as soil organic fertilizers, Cu and Zn represent the most critical HMs in intensive pig production. Sustainable approaches that consider both input and output HMs are urgently needed to guarantee the reduction of the environmental pollution from livestock-related activities. Firstly, levels of Cu and Zn in diets for growing pigs should be reduced without detrimental effects on the production and mineral status. Secondly, higher dietary bioavailable organic complexes of these metals lead to a substantial reduction in the dietary inclusion rate, which should have a positive outcome for pig health and environmental sustainability. Thirdly, the potential sources of HMs outputs from livestock wastewater to the environment should be controlled. Integrated plant-based strategies such as CWs are thus a valuable tool for phytoremediation in order to reduce the high content metals from livestock wastewater.

Constructed wetlands are largely used for pollutant recovery of wastewater from different sources, and their efficiency of CWs in recovering HMs critically depends on the differences in uptake and translocation of HMs and other pollutants among plants used for phytoremediation. However, it is difficult to quantitatively define the performance of plants since the environment created in the CWs heavily shapes the pollutant removal efficiencies. Yadav et al. [89] showed that the wetland bed depth has direct significant effect on HMs removal efficiencies in vertical flow CWs. In fact, the removal of Cr, Ni, Cu, Zn and Co increased by 16.6%, 22.9%, 20.4%, 21.5%, and 21.8%, respectively, when the gravel bed depth of CWs was increased from 0.3 to 1.5 m. In addition, the presence of various microorganisms and the initial concentration of HMs also affect plant performance [127]. The higher HMs concentration in water induces the higher uptake by plants [95]. The pattern of CWs is also critical in conditioning the efficiency of plants in pollutant uptake. Sandoval et al. [104] presented a synthesis which could be used in the design of new CWs and suggested “there is no clear pattern in the use of a specific plant species for a certain type of wastewater”, thus making it difficult to associate specific plants with a specific pollutant uptake. Compared to chemical and physical approaches, phytoremediation is thus more effective in counteracting Zn, Cu, and other sources of metal pollution. It also offers new means for metal recovery, leading to innovative high-value raw materials and valuable organic compounds.

Despite the potential of phytoremediation to result in hazardous biomass, the application of a proper downstream processing of the biomass can transform waste into a high-value material. If the downstream processing of the exhausted harvested plants is carefully designed, landfilling of the biomass itself or even of its ashes can be prevented, thus contributing to its benefits. Several downstream approaches are possible, and between other incineration and anaerobic digestion, are probably those of choice, considering the amount of waste to be treated. After phytoremediation, biomass can be treated thermochemically, through gasification, pyrolysis, or combustion, thereby providing fuel gas that can be used for electricity generation or to produce heat and a reduced volume of the ashes compared to the volume of the biomass. Heat generated directly through combustion or from the fuel gas can be used to foster plant growth and for the supply of ancillary items of the phytoremediation plant. The resulting ash could also be used as a pozzolanic addition to hydraulic binders, in the formulation of composite Portland cement and geopolymers. The production of biogas by anaerobic fermentation of the plant biomass used for the phytoremediation is also a possible circular approach. The biogas mixture obtained on average contains 60% methane, which can be used as a substitute for fuel in boilers. As it has high N, P, and K contents, the associated liquid fraction could also be used in agriculture. Combining phytoremediation and biorefinery could therefore be developed into a sustainable strategy which would add value to both approaches, enabling metal recovery and producing valuable sugars and organic compounds.

In conclusion, in order to move toward a more resource efficient and sustainable food system, it is essential to find more efficient ways to improve the technical knowledge on the environmental impacts of food, stimulating sustainable livestock production. Regarding soil and water quality, livestock production systems have the highest impact on agricultural pollution, particularly in terms of the animal-manure management by farms. There is currently a great interest in new ways to manage the water contamination and manure management within farms.

## Figures and Tables

**Figure 1 ijerph-18-02239-f001:**
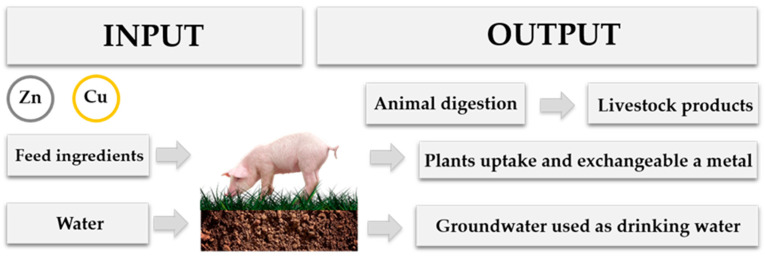
Possible routes of HMs entrance to the food chain and the consequences of their output.

**Figure 2 ijerph-18-02239-f002:**
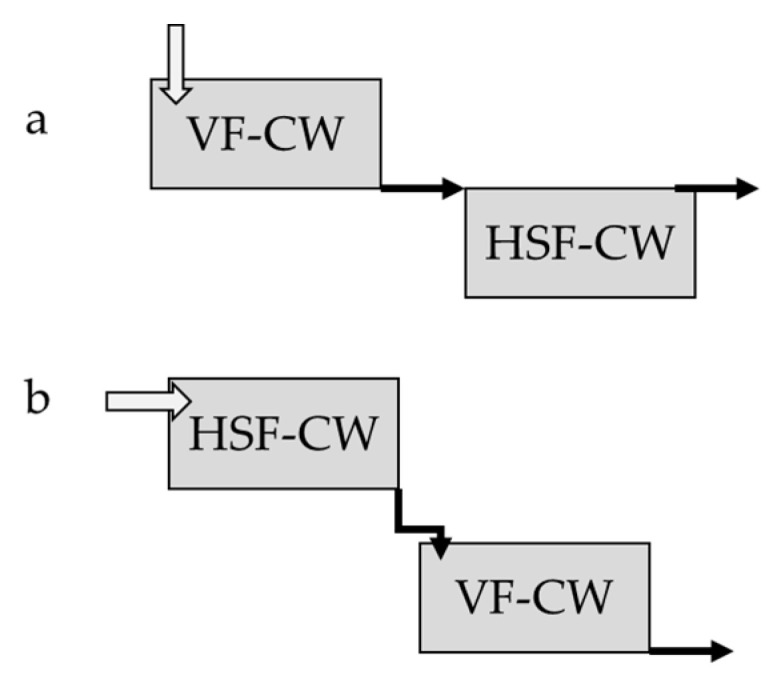
Diagram of hybrid constructed wetland where, (**a**) is vertical flow SF-CWs followed by horizontal flow SF-CWs and (**b**) is horizontal flow SF-CWs followed by vertical flow SF-CWs.

**Table 1 ijerph-18-02239-t001:** Heavy metals in animal nutrition [6,9].

Essential Elements (authorized in animal nutrition according to EC N°1831/2003)								
Co (cobalt)	Cr (chromium)	Cu (copper)	Fe (iron)	Mn (manganese)	Mo (molybdenum)	Ni (nickel)	Se (selenium)	Zn (zinc)
**Nonessential elements** **(undesirable elements according to 2002/32/EC)**								
As (arsenic)			Cd (cadmium)		Hg (mercury)		Pb (lead)	

**Table 2 ijerph-18-02239-t002:** Concentration of Zn and Cu in livestock manures [27,28,29,30].

Area	Heavy Metal			Source of Heavy Metals		
				Swine Slurry	Cattle Slurry	Poultry Slurry
England	Zn	mg/kg d.w.		650.0	170.0	217.0
	Cu			470.0	45.0	32.0
Netherlands	Zn	mg·kg^−1^		186.2	73.7	-
	Cu			644.7	296.3	-
China	Zn	mg/kg d.w.		843.3	151.9	308.9
	Cu			472.6	46.5	102.0
China	Zn	mg/kg d.w.	^a^ S	119.1	674.7	268.2
			^b^ M	126.3	476.0	241.7
			^c^ L	136.1	691.6	384.2
	Cu		S	30.8	958.8	51.6
			M	31.0	420.4	57.2
			L	31.4	612.2	87.1

^a^ S—small animal population (head): cattle <100, chicken <2000, swine <200. ^b^ M—middle animal population (head): cattle 100–300, chicken >2000, swine 200–800. ^c^ L—large animal population (head): cattle >300, chicken >20,000, swine >800. d.w.—dry weight.

**Table 3 ijerph-18-02239-t003:** Annual input of Zn and Cu in soil for 1 mln of ha [27,28,34].

Region		China	France	Germany	United Kingdom	Netherlands
Total land area (mln ha)		122.0	29.0	17.0	11.1	2.0
Heavy metals	Zn (g/ha^−1^)	1538.9	523.8	1249.2	453.9	684.5
	Cu (g/ha^−1^)	588.7	167.9	269.2	146.0	294.0

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
