# Peer review of "Heavy-Metal Phytoremediation from Livestock Wastewater and Exploitation of Exhausted Biomass"

_ijerph, 2021, doi:10.3390/ijerph18052239_

Round 1

Reviewer 1 Report

The Authors The heavy metal phytoremediation from livestock wastewater and valorization of exhausted biomass is shown in this paper.
It's a very interesting study.
The Authors note that the coupling of phytoremediation and biorefinery processes could thus provide a suitable strategy, which would add value to both approaches and facilitate metal recovery.
This review focuses on the concept of agro-ecology with the emphasis on the excessive use of heavy metals in animal production, the different techniques and adaptations of the heavy metal phytoremediation from livestock wastewater and further applications of exhausted phytoremediated biomass.
There is currently a huge interest in new ways for managing the water contamination and manure management within farm.
The Authors suggest that in order to move towards a more resource efficient and sustainable food system, is mandatory to find more efficiency way to better technical knowledge on the environmental impacts of food, stimulating sustainable livestock production.
The references were well matched to the research topic.
I believe this paper doesn't need any further revision. I approve of the consideration for publication in current form.

Author Response

Dear referee,

The authors thank the reviewer for their previous comments and suggestions that improved the overall quality of the paper. Sincerely yours

Reviewer 2 Report

I have carefully reviewed the revised version of “Heavy Metal Phytoremediation from Livestock Wastewater and Valorization of Exhausted Biomass”. I am happy to see that the quality of manuscript has been improved. However, the chapter 7 should be re-edited. 

I don’t think all the argument about phytoremediation of heavy metal and disposal of exhausted biomass is necessary in Chapter 7.

I think that the title of Chapter 7 should be “Conclusion and future directions”. This chapter is required to rewrite.

L652-656, The first paragraph can be condensed into one sentence.

For chapter 7, some parts of argument can be transferred to the Chapters 2 - 6.

Author Response

Dear referee,

The authors would like to thank you for your comments and suggestions that were intended to improve the overall quality of the paper. We have completely revised and improved further the manuscript, taking into account your suggestions.

  • I don’t think all the argument about phytoremediation of heavy metal and disposal of exhausted biomass is necessary in Chapter 7.
  • I think that the title of Chapter 7 should be “Conclusion and future directions”. This chapter is required to rewrite.
  • For chapter 7, some parts of argument can be transferred to the Chapters 2 - 6.

In the previous version, the content of chapter 7 was discussed in other chapters. However, referee 1, in the first revision, asked us to create a new chapter (Discussion and conclusions, Chapter 7.). Modifications required by Referee 2 would abolish most of changes we have done to answer of comments of referee 1 in previous revision. Since referee 1 wrote: "I believe this paper doesn't need any further revision. I approve of the consideration for publication in current form", we think that it would not be suitable to rewrite sections 6 and 7.

We inserted minimal modifications without changing what previously asked by referee 1.

  • Sentence was moved from pg. 9 lanes 386-387 to pg 3 lanes: 101-103.
  • Sentence was moved from pg. 9 lanes 431-433 to pg 8 lanes: 362-363.

 L652-656, The first paragraph can be condensed into one sentence.

The lanes correspond to references. However, we changed the sentences Pg 9 lanes 394-397.

 We hope that this revision satisfies your request completely and we should be grateful if you would consider the manuscript for IJERPH

Sincerely yours

Reviewer 3 Report

The paper especially as one proceeds still is lacking in clarity of sentences. 

Additionally there are in this revised version several places where a statement seems contradicted by a subsequent statement.

Please find a qualified editor who can work with the science for your paper. 

Author Response

Dear referee,

We are not able to answer since the referee made general comments and does not indicate specific points.

 Sincerely yours

Reviewer 4 Report

This review presents a very important point related to sustainable agriculture: the need to take care about heavy metal pollution (mainly Zn and Cu) associated to animal production. Authors explain extensively the origin of the problem and possible new strategies for eliminating heavy metals from soil and water. Specifically, they suggest a novel strategy where phytoremediation and biorefinery are coupled. The manuscript is properly written and all the concepts are clearly explained, being very easy to read and understand it.

From my point of view, I recommend to accept the manuscript for its publication in its present form. I only have some minor questions/comments for authors:

  • Does the use of zinc and copper as an alternative to antibiotics have any negative impact on pig’s health? Are these heavy metals then released to the environment and accumulated as toxic forms in the ecosystems? I suppose that they would reach high concentrations if the use of these heavy metals is so generalized. Some numerical data related to this point could be added to the text (Lines 76-194).

  • I do not really understand the association mentioned in the text (Lines 210-213) related to the emergence of methicillin-resistant Staphylococcus aureus and the increased use of copper or zinc as alternative antibiotics. Maybe a sentence clarifying this point may be add to the text.

  • How could be explained that in the case of palustris and T. latifolia a significant reduction of their cell wall thickening is observed instead of the opposite effect that is normally observed? Have they developed different strategies for being tolerant to heavy metals? (Lines 513-516). Some short clarification could be added to the text related to this point.

Author Response

Dear referee,

The authors would like to thank you for your comments and suggestions that were intended to improve the overall quality of the paper. We have completely revised and improved further the manuscript, taking into account your suggestions.

  • Does the use of zinc and copper as an alternative to antibiotics have any negative impact on pig’s health? Are these heavy metals then released to the environment and accumulated as toxic forms in the ecosystems? I suppose that they would reach high concentrations if the use of these heavy metals is so generalized. Some numerical data related to this point could be added to the text (Lines 76-194).

Firstly, we would like to say, that we explained if the Zn and Cu have negative impact on pigs’ health in the lines 61-62 and 89-93. Moreover, the HMs are released via animal manure and accumulated in the agricultural soil as toxic elements. We discussed extensively about HMs in animal feeds, manure, soil and animal origin product in the previous review of Hejna et al. (2018) and we only added the citation of this review in this chapter. However, we added few sentences to introduce the subject and we added numerical data related to Zn and Cu content in animal manure and agricultural soil following the review of Hejna et al (2018). Please see Pg 3,4; lanes 108-126 Pg 4, lanes 151-153.

  • I do not really understand the association mentioned in the text (Lines 210-213) related to the emergence of methicillin-resistant Staphylococcus aureus and the increased use of copper or zinc as alternative antibiotics. Maybe a sentence clarifying this point may be add to the text.

We should have been explained this issue in proper way. Thus, we cited the review of Yazdankhah et al. 2014, where authors showed many citations where the use of Zn and Cu in the diet might create possible antimicrobial resistance. Moreover, we clarified this aspect in the current version. Please see the Pg. 3, lanes 127-141.

  • How could be explained that in the case of palustris and T. latifolia a significant reduction of their cell wall thickening is observed instead of the opposite effect that is normally observed? Have they developed different strategies for being tolerant to heavy metals? (Lines 513-516). Some short clarification could be added to the text related to this point.

We insert a sentence to clarify this point (Pg 7, lanes 294-298). The main discussion of this point was reported in the paper 50.

We hope that this revision satisfies your request completely and we should be grateful if you would consider the manuscript for IJERPH

Sincerely yours

Round 2

Reviewer 2 Report

I have carefully reviewed the revised version of "Heavy Metal Phytoremediation from Livestock Wastewaters and Valorization of Exhausted Biomass" . The authors have edited this manuscript well according to suggestions and comments.Therefore, I suggest the acceptance of this revised manuscript in plant growth regulation. Best regards, Ping

Author Response

The authors thank the reviewer for their previous comments and suggestions that improved the overall quality of the paper.

Sincerely yours

Elisabetta Onelli

Reviewer 3 Report

still some suggested changes  especially creation of a table that shows the levels of HM contamination in sludges      it would aid the reader  

Author Response

Dear referee,

The authors would like to thank you for your comments and suggestions that were intended to improve the overall quality of the paper. We have completely revised and improved further the manuscript, taking into account your suggestions.

The English editing was performed.

 Comment: Pg 3 lane 112

The tables were inserted.

  • Comment: Pg 6 lane 236

A diagram was inserted.

  • Comment: Pg 7 lane 282

We report a part of discussion to answer to this comment in the paper Stroppa et al., [50]: The accumulation of starch in the leaves could be due to changes in carbohydrate metabolism. In fact, in several plants, metal exposure causes a variation in photosynthetic process accompanied by significant alterations in plant biomass and leaf morphology/ultrastructure (Arif et al, 2016; Rufner and Barker, 1984; Stoláriková-Vaculíková et al., 2015; Todeschini et al, 2011). One of the mechanisms of metal tolerance has been shown to be the accumulation of metals in the ageing leaves (MaÅ‚achowska-Jutsz and Gnida 2015). The higher accumulation of amyloplasts in older with respect to younger leaves, also suggested that in T. palustris, such stress avoidance may occur. In most plants, high copper exposure disorganizes the chloroplast ultrastructure without starch accumulation (Maksymiec et al., 1996), while in screwbean mesquite, excess copper affects chloroplast development leading to starch accumulation in cotyledons (Zappala et al, 2014). An alternative hypothesis for the presence of amyloplasts in leaves could be due to a modification of carbohydrate translocation away from leaves. This hypothesis was supported by the decrease in starch granules in the rhizomes in treated plants with respect to the control, as also confirmed by chemical analyses (see accompanying paper, Heina et al., 2019). In screwbean mesquite, the increase in copper was accompanied by a decrease in potassium concentration (Zappala et al, 2014). Interestingly, the alteration of Na and K homeostasis, disturbed phloem loading and translocation, leading to an accumulation of starch in Arabidopsis leaves (Tian et al, 2010).

  • Comment: Pg 7 lane 321

The sentences were modified as follow: During phytoremediation, the pollutants are concentrated in the plants, and ashing of the exhaust harvested plants further concentrates the metals, making recycling possible. Moreover, the recovery of the ashing generated heat may be used for enhancing plant growth, [106,107]. Alternatively, the plants biomass may be used for biogas production in anaerobic fermentation processes [108-110]. In the design of a phytoremediation approach, the downstream processing of the exhausted plants, when properly planned, could prevent direct landfilling of the biomass or even of ashes, but also it could even result in a gain deriving from the revaluation of these waste.

  • Comment: Pg 8 lane 346

The reference was inserted and the sentences were modified as follow: Metal enrichment in the ashes could also impact on metal recovery and recycling of metals, in particular those included in the critical raw materials list [115] Recovery has great potential when a limited number of metals are present in relatively high concentrations. Among others, examples are Cu and Zn, present in swine wastewater, as a result of their addition to the animal feed for their antibacterial and anti-inflammatory activities. Another example is the recovery and recycling of the nickel, whose demand is increasing for the production of batteries, to replace cobalt both for political and environmental issues [116]

We hope that this revision satisfies your request completely and we should be grateful if you would consider the manuscript for IJERPH

Sincerely yours

Elisabetta Onelli

This manuscript is a resubmission of an earlier submission. The following is a list of the peer review reports and author responses from that submission.

Round 1

Reviewer 1 Report

The informations review of the heavy metal phytoremediation from livestock wastewaters and valorization of exhausted biomass is shown in this paper. This review focuses on the concept of agro-ecology with the emphasis on the excessive use of heavy metals in animal production, the different techniques and adaptations of the heavy metal phytoremediation from livestock wastewater and further applications of exhausted phytoremediated biomass.
The Authors note that at although zinc and copper are essential nutrients with positive effects on growth performances their diffusion into the environment should be controlled. Moreover, it is important to consider the risk and benefits in order to reduce the input and output of these elements in animal production systems.
The Authors suggest that integrated approaches in line with circular economy principles are needed which could thus increase the sustainability of animal production and they note that phytoremediation is more effective in counteracting Zn, Cu and other metal pollution compared to chemical and physical approaches.
References have been well selected the content of the paper.

Some suggestions follows:
- It must be to rewrite the Conclusions chapter to for better understand this issue. Moreover, the Conclusions chapter should wider summarize the analysis of previously cited information from the references. Or... it should be good to add Discussion chapter with wider analysis of references or combine a Discussion chapter with added Conclusion chapter (as Discussion and Conclusion chapter).
- Please check spelling of "wastewaters", I think that it should be "wastewater".
- I should be good to expand the number of references.

- In the description on the MDPI website, there is information about 27 pages of this review, but in fact the review have a 15 pages. Is this an error in the description?

Author Response

Dear referee,

The authors would like to thank you for your comments and suggestions that were intended to improve the overall quality of the paper. We have completely revised and improved further the manuscript, taking into account your suggestions.

Below you can find the responses, item by item, related to the paper in the object.

 1) “It must be to rewrite the Conclusions chapter to for better understand this issue. Moreover, the Conclusions chapter should wider summarize the analysis of previously cited information from the references. Or... it should be good to add Discussion chapter with wider analysis of references or combine a Discussion chapter with added Conclusion chapter (as Discussion and Conclusion chapter).”

We rewrite and expand the Conclusion as suggested by Referee (Pg 8-10) creating a new chapter called “Discussion and conclusion”.

2) Please check spelling of "wastewaters", I think that it should be "wastewater".

We changed "wastewaters" with "wastewater" in all manuscript.

3) I should be good to expand the number of references.

The number of references was expanded from 104 to 121. The references in the text were revised.

4) In the description on the MDPI website, there is information about 27 pages of this review, but in fact the review have a 15 pages. Is this an error in the description?

Actually, pages of manuscript were 15. We are sorry for the mistake.

We hope that this revision satisfies your request completely and we should be grateful if you would consider the manuscript for IJERPH

Sincerely yours

Elisabetta Onelli

Reviewer 2 Report

Comments to Author:

Title: Heavy Metal Phytoremediation from Livestock Wastewaters and Valorization of Exhausted Biomass

With the rapid development of large-scale livestock and poultry breeding industry, a large number of livestock and poultry feces and urine from feed additives and veterinary drug residues are produced, which makes the problems heavy metals pollution in aquaculture wastewater increasingly prominent. The long-term discharge of livestock wastewater leads to the enrichment of heavy metals in land and river. Phytodepuration is a low cost and ecologically friendly technology for civil and industrial wastewater refinement. This paper focuses on the concept of agro-ecology with the emphasis on the excessive use of heavy metals in animal production, the different techniques and adaptations of the heavy metal phytoremediation from livestock wastewater and further applications of exhausted phytoremediated biomass. To construct a green and efficient aquaculture wastewater treatment system, it is very necessary to provide ideas and reference for the future exploration of heavy metal pollution wastewater remediation technology.

The overall idea of the article was written clearly. However, there are some improvement required to clarify.

Specific comments:

  1. In section 3: Since livestock and poultry breeding has caused pollution of Cu and Zn. Due to Zn2+ and Cu2+ are the most critical output from swine livestock, it is very important to provide some detail data to explain the Cu and Zn pollution degree of wastewater.
  2. In section 4, author suggested that plants could be successfully employed in in situ phytoremediation systems, to remove Cu2+ and Zn2+. Which plants have the potential to be used in situ phytoremediation of livestock? Please provide a list to explain.
  3. In section 4.2, CWs are widely used for HM remediation or other organic pollutants. It is very necessary to draw a working principle flow chart of CWs to explain the treatment process of pollutants in aquaculture wastewater by constructed wetland. The authors have enumerated many plants that are capable of purification. Can you compare their functions in tabular form? Please provide them.

Author Response

Dear referee,

The authors would like to thank you for your comments and suggestions that were intended to improve the overall quality of the paper. We have completely revised and improved further the manuscript, taking into account your suggestions.

1) In section 3: Since livestock and poultry breeding has caused pollution of Cu and Zn. Due to Zn2+ and Cu2+ are the most critical output from swine livestock, it is very important to provide some detail data to explain the Cu and Zn pollution degree of wastewater.

We added the detailed citations about research studies related to the content of Zn and Cu in feed, manure and wastewater. Please see the pg. 3, lanes 86-101. In addition, a new chapter (4. Heavy metals and their impact on the environment) highlights in more detail the contamination of manure and wastewater by Cu and Zn.

2) In section 4, author suggested that plants could be successfully employed in in situ phytoremediation systems, to remove Cu2+ and Zn2+. Which plants have the potential to be used in situ phytoremediation of livestock? Please provide a list to explain.

For phytoremediation of livestock, macrophytes are the most used plants, probably thanks to their tolerance to HMs exposure, the high underground biomass able to accumulate HMs and the low ability to translocate them in shoot. In addition, they are adapted to submerse or semi-submerse environment making them useful for each CW types. We inserted sentences in the text pg. 5, lanes: 212-215 and pg. 5-6, lanes 227-233.

3) In section 4.2, CWs are widely used for HM remediation or other organic pollutants. It is very necessary to draw a working principle flow chart of CWs to explain the treatment process of pollutants in aquaculture wastewater by constructed wetland. The authors have enumerated many plants that are capable of purification. Can you compare their functions in tabular form? Please provide them.

I agree with referee that a table reporting functions of different plants used in the CWs could be a useful instrument for appropriate CW design. However, as reported in literature, many parameters affect the efficiency of the same plant in CWs, as for example the depth, the type of CWs, the co-presence of different plants in the same macrocosms, the nature of substrate and so on. Therefore, it is very difficult to create this table. However, a synthesis has been already made in a recent paper by Sandoval et al. 2019 (Sandoval, L.; Zamora-Castro, S.A.; Vidal-Álvarez, M.; Marín-Muñiz, J.L. Role of Wetland Plants and Use of Ornamental Flowering Plants in Constructed Wetlands for Wastewater Treatment: A Review. Appl. Sci. 2019, 9, 685). Moreover, we inserted sentences in the text to explain this aspect. Please see pg 6 lanes 271-276 and pg 9 lanes 398-411.

We hope that this revision satisfies your request completely and we should be grateful if you would consider the manuscript for IJERPH

Sincerely yours

Elisabetta Onelli

Reviewer 3 Report

The review has a minimally limited scope and does not offer any scientific significance to the field. Many broad claims have been repeatedly stated without any knowledge depth.

Author Response

Dear referee,

The authors would like to thank you for your comments and suggestions that were intended to improve the overall quality of the paper. We have completely revised and improved further the manuscript, taking into account your suggestions.

  • The review has a minimally limited scope and does not offer any scientific significance to the field. Many broad claims have been repeatedly stated without any knowledge depth.

The review focused on the use of heavy metal phytoremediation for livestock wastewater reuse and valorization of exhausted biomass. In our best knowledge, no review currently explained in good manner the connection among the broad spectrum of animal nutrition and health, environmental pollution control and applications of exhausted biomass, in line with circular economy principles. In any case, we particularly improved chapters (1-4) and we added more citations. Therefore, in our opinion, the review has interesting and novel aim and thus offers a broad scientific significance to the field.

Furthermore, the manuscript has been English revised before the submission by a mother tongue referee (English for Academics di Adrian John Wallwork company). Please see the uploaded certificate of English revision.

We hope that this revision satisfies your request completely and we should be grateful if you would consider the manuscript for IJERPH

Sincerely yours

Elisabetta Onelli

Reviewer 4 Report

The authors have presented a review that is easy and valuable to read-- maybe the answer to the pig raising is effective probiotics   and more human conditions 

I have made comments as sticky notes  -  in a few places I cannot understand the flow of what is being said and those require fixing.  I think the addition of more sentences to eliminate the leaps in thought is needed,

in other cases i cannot see based on where we are in the world that the method is feasible without  extensive interdisciplinary and infrastructure planning 

but raising issues and getting discussion going is needed  

Author Response

Dear referee,

The authors would like to thank you for your comments and suggestions that were intended to improve the overall quality of the paper. We have completely revised and improved further the manuscript, taking into account your suggestions.

  • I have made comments as sticky notes -  in a few places I cannot understand the flow of what is being said and those require fixing.  I think the addition of more sentences to eliminate the leaps in thought is needed, in other cases i cannot see based on where we are in the world that the method is feasible without  extensive interdisciplinary and infrastructure planning but raising issues and getting discussion going is needed.

We changed chapters 1-3 and we added new chapter 4 in order to connect better the ideas and make it more readable without unnecessary leaps. We also insert some sentences to better connect chapters.

We hope that this revision satisfies your request completely and we should be grateful if you would consider the manuscript for IJERPH

Sincerely yours

Elisabetta Onelli
